# Digital Twin-Driven Rear Axle Assembly Torque Prediction and Online Control

**DOI:** 10.3390/s22197282

**Published:** 2022-09-26

**Authors:** Lilan Liu, Zifeng Xu, Chaojia Gao, Tingting Zhang, Zenggui Gao

**Affiliations:** 1School of Mechatronic Engineering and Automation, Shanghai University, Shanghai 200444, China; 2Shanghai Key Laboratory of Intelligent Manufacturing and Robotics, Shanghai University, Shanghai 200444, China

**Keywords:** digital twin, gray wolf optimized variational modal decomposition, long short-term memory networks, torque prediction, online control

## Abstract

During the assembly process of the rear axle, the assembly quality and assembly efficiency decrease due to the accumulation errors of rear axle assembly torque. To deal with the problem, we proposed a rear axle assembly torque online control method based on digital twin. First, the gray wolf-based optimization variational modal decomposition and long short-term memory network (GWO-VMD-LSTM) algorithm was raised to predict the assembly torque of the rear axle, which solves the shortcomings of unpredictable non-stationarity and nonlinear assembly torque, and the prediction accuracy reaches 99.49% according to the experimental results. Next, the evaluation indexes of support vector machine (SVM), recurrent neural network (RNN), LSTM, and SVM, RNN, and LSTM based on gray wolf optimized variational modal decomposition (GWO-VMD) were compared, and the performance of the GWO-VMD-LSTM is the best. For the purpose of solving the insufficient information interaction capability problem of the assembly line, we developed a digital twin system for the rear axle assembly line to realize the visualization and monitoring of the assembly process. Finally, the assembly torque prediction model is coupled with the digital twin system to realize real-time prediction and online control of assembly torque, and the experimental testing manifests that the response time of the system is about 1 s. Consequently, the digital twin-based rear axle assembly torque prediction and online control method can significantly improve the assembly quality and assembly efficiency, which is of great significance to promote the construction of intelligent production line.

## 1. Introduction

The bolt connection is one of the most widely used connection methods in the rear axle assembly, which is a key part of the automobile manufacturing process [1]. By applying a certain torque on the bolt pair, the connecting pieces are fixed to each other, so the design requirements can be met. Due to simple methods, cost-effective and easy supervising, torque control method [2] has been commonly used for automobile assembly, which is directly related to vehicle quality and operational reliability. Many factors, such as the material, diameter, and surface roughness of threads, the surface friction coefficient of bolts and connecting parts, the precision of tightening tools, rotation speeds, and the sequence of the tightening process, have a significant influence on the final torque result [3]. Due to the factors of the vibration of tightening guns, fixture wear, the plastic deformation of workpiece materials, and the accumulation of positional deviation, the output torque of tightening guns cannot meet the current assembly requirements, affecting the assembly quality of the rear axle. Therefore, an effective assembly torque prediction and control method is required to ensure the stability of the rear axle assembly. Furthermore, high-precision assembly torque prediction models and control methods can avoid high-cost and long-term measurement processes. Many researchers have tried to develop the prediction methods of bolt assembly torque, and the models mainly include mechanism model methods and artificial intelligence methods [4,5,6].

The torque prediction method based on the mechanism model is to establish torque prediction mathematical model by considering fixture shape, material properties, assembly error, contact mechanics, and other factors. According to the difference in tightening torque of bolts, Kang et al. [7] put forward a contact mechanics algorithm to predict the bolt bottom torque by the parameters of surface topography, root-mean-square height, and hardness. Using Physical Experiment and Finite Element Simulation Method, Ekrem Oezkaya et al. [8] developed a torque prediction method based on a combination of geometric and mechanism models to predict torque values during automatic tapping. According to the variation of SiCp/Al6063 material properties and the relation between torque and tool life, Teng Dou [9] presented a finite element constitutive model for composites to predict thrust and torque during drilling. Zhen Zhang et al. [10] proposed a linear acoustic method based on wave energy dissipation (WED), and used the Hertz contact theory model to predict the loosening torque of bolts. Fengrui Liu [11] studied the influence of the randomness of the opening strength and characteristics of the laminate, geometric parameters, and clearance on the bolt torque, and proposed a composite multi-bolt tightening torque prediction method. Eiji Shamoto [12] proposed a torque prediction in bolted connection. By studying the torsional contact model of connecting bolts and utilizing finite element analysis to extract and solve linear phenomena, the author realized the prediction of contact parameters and torque. T. Dang Hoang [13] used Latham and Crockoft’s energy model to perform calculations in elastoplastic with/without damage, and considering the tightening torque and clearance parameters, used a finite element simulation method to predict the stress relaxation of bolted connections.

In torque prediction methods, due to the low parameter identification of the mechanism model, the bolt assembly torque cannot be accurately predicted. Therefore, many researchers have adopted artificial intelligence methods to predict assembly torque based on historical and real-time data. Ali Djerioui [14] proposed a torque predictive control (PTC) method for permanent magnet synchronous motor (PMSM), by improved gray wolf optimization algorithm (GWO). Kairong Hong [15] studied some key parameters such as torque, and established the relation model between torque and thrust of cutter head based on long short-term memory (LSTM) network to realize the forecast of shield machine cutter head torque. Shi Gang [16] used variational mode decomposition (VMD) to decompose the cutterhead torque of shield machine, and established a multi-step forecast model to predict the cutterhead torque of shield machine. Chengjin Qin [17] developed a multi-step forecast model of cutterhead torque based on adaptive decomposition method by using the original cutterhead torque data as input. X. Xia [18] realized the prediction of bearing torque by using improved neural network first-order weight coefficients and Volterra series. According to the experimental results, the five prediction models have small prediction errors and high accuracy advantages. Mahdi Bagheripoor [19] used finite element simulation method to obtain process parameters, and used artificial intelligence methods to forecast the rolling force and torque of a hot rolling mill. Weiying Zeng [20] proposed a neural network engine torque method, by using a single hidden-layer neural network structure, and this method effectively improves the prediction accuracy. Zhang et al. [21] extracted features from the shield machine torque feature data and uses a hybrid multi-model approach to predict torque. 

Due to the dynamic change of assembly torque, manufacturing enterprises urgently need to realize the online control of assembly torque on the production line under the premise of obtaining the prediction value of assembly torque, and combine the historical data and real-time data to supervise and visualize the status of the production line [22]. In 2002, Grieves [23] proposed the concept of digital twins, describing the relationship between physical entities and virtual entities. In previous studies, Qinglin Qiet al. [24] studied the basic model and method of digital twin, and established the technical route of digital twin, which provided technical support for its application. Fei Tao et al. [25] raised a five-dimensional model of a digital twin workshop, which provides the mechanism and data support for the health prediction of production line equipment [26,27,28]. Tao et al. [29] proposed the basic application of digital twin in the workshop, and introduced the workshop information interaction method in detail, and it provided support for the construction of a digital twin production line. Fei Shen [30] introduced digital twin technology in the oil and gas production process, realized the visualization of oil and gas production process, and monitored the status of production equipment, which improved the safety and production efficiency of oil and gas production. Li Shi [31] developed the digital twin system of the aerospace assembly workshop, integrated the workshop’s logistics information, product information, production scheduling information, etc., and optimized the workshop production efficiency to achieve precise production control of the production line. Xin Tong et al. [32] added sensing equipment to the machine tool, collected data during the production process of the base machine tool, and realize the visualization of the machine tool production process and the monitoring of machining status by a machine tool digital twin system. Liu et al. [33] raised a product quality monitoring method by digital twin technology, and the product quality was monitored and the production process was controlled by collecting and analyzing the relevant data of the production line. Zengya Zhao [34] proposed a feedback control method for workpiece surface roughness prediction and adaptive surface roughness stability through digital twin technology.

With the wide application of six-axis robotic arms in production, it has become a common way for the rear axle assembly process to use the robotic arm to drive the tightening gun to output assembly torque [35]. However, due to the accumulation of errors in the attitude, output angle, and other factors of the tightening gun during the assembly process, the assembly torque cannot meet the quality requirements of the body, and the assembly efficiency is affected. In addition, the production line assembly torque fluctuations, resulting in the assembly workpiece not satisfying the quality requirements, increasing the assembly time and cost. Therefore, an effective method is needed to realize assembly torque prediction and control. From the above literature research, the current methods for parameter prediction are mainly based on physical model and data-driven methods. The physical model prediction method has poor prediction accuracy due to low parameter identification, while the data-driven method has high prediction accuracy for stationary data and poor prediction effect for non-stationary data. In the face of dynamic and random assembly torque, only the method of directly predicting the original torque data is obviously less effective. The VMD decomposition algorithm “decomposes-reconstructs” the original data, which can convert non-stationary data into stationary data, reduce data complexity and improve the accuracy of assembly torque trend prediction. Aiming at the high complexity and high dynamic characteristics of rear axle assembly, a torque prediction and on-line control method of assembly bolts digital twin model is raised by using the characteristics of virtual–real interaction of digital twin. Therefore, the specific problems solved in this paper are as follows: (i) Aiming at the non-stationary characteristics of the original assembly torque, the assembly torque prediction based on ‘decomposition-reconstruction‘ and an artificial intelligence method was proposed. (ii) Aiming at the problem of insufficient information interaction in assembly line, the digital twin system of rear axle assembly line was developed to realize the visualization and supervising of assembly process. (iii) The torque prediction module was coupled to the digital twin system for real-time prediction and online feedback control of assembly torque. The article structure is divided into seven parts, the organization of the other sections is listed as follows: In Section 2, we build the general architecture of the proposed method. In Section 3, the digital twin rear axle assembly line model was established. In Section 4, a variational modal long short-term memory network GWO-VMD-LSTM torque prediction algorithm based on gray wolf optimization is proposed. In Section 5, testing and comparisons of torque prediction algorithms. In Section 6, the method combining the proposed predictive model with digital twin realize an online control of assembly torque. Finally, the conclusions and future work are given.

## 2. Overall System Architecture

With the purpose of realizing the prediction and online control of rear axle assembly torque, we needed to establish a real-time data acquisition system. The collected data was transferred to the data storage platform after processing, and the assembly process was supervised by the digital twin system. By using the collected data, the real-time prediction of assembly torque was realized. Finally, the online control of assembly torque was realized through the decision module. Figure 1 shows the overall architecture of the rear axle assembly line digital twin system, including three modules:

Data acquisition module: This module acquires the real-time data of the physical rear axle assembly line. The collected data includes: status data, torsion angle, torque data, etc.Data processing module: This module realizes the function of data filtering, data modeling, data fusion, data analysis, and data storage. The collected data will be processed to reduce data redundancy and ensure validity. Then, the data was stored in a structured way after data analysis and fusion to make system development convenient.Service module: The specific aspects of application services based on digital twin are as follows: (i) equipment state supervising and visualization of the assembly process, (ii) assembly torque prediction, and (iii) feedback control of the assembly process. Firstly, virtual model of the assembly line was established, and the virtual assembly line was driven by the real-time data, so the supervising of assembly parameters and visualization of the assembly process were realized. Then, a torque prediction algorithm model was proposed by analyzing the data of the assembly line to provide intelligent decision-making for the assembly line. Finally, the control command is generated to realize the online control of assembly torque.

## 3. Build Digital Twin Rear Axle Assembly Line Model

### 3.1. Twin Element Modeling

The virtual rear axle assembly line must reflect all the characteristics of the real assembly line, the virtual model, and reflect the production status of the real assembly line. The virtual production line needs to be constructed from the perspectives of production products, production equipment, and production environment. Therefore, the specific factors to be considered are as follows:Products twin modelingAt different stages of the process, different models of products correspond to different parts, accompanied by order, quality, coding, and other full-lifecycle information, which can be stored in the virtual label of each product in the digital space using data interfaces, and at the same time drive the evolution of the product’s component composition according to its process information. Equipment twin modelingFor the purpose of realizing the mapping of the twin model to the physical entity, the equipment model must be ensured that the three-dimensional size information and behavior rules are highly consistent with the physical entity. The twin model needs to establish a virtual–real communication transmission interface to obtain real-time data from the device, and define relevant virtual behavior rules to realize data-driven model behavior. The twin modeling of assembly line environmentThe twin modeling of production environment mainly includes the field lighting of assembly line, the main production equipment material, texture, mapping, auxiliary equipment construction, factory building, etc., rendering the real production environment to improve the visual fidelity of twin model. In addition, the environmental temperature, humidity, and other data collected by the sensors need to be applied to the virtual scene to ensure high consistency between virtual and reality.

### 3.2. Virtual–Real Mapping

The relationship between data and model is increasingly important to construct the digital twin system. In the collected data, the command data is used for the starting and ending behavior of the model, and the state data is connected to the state behavior of the model. As shown in Figure 2, the mapping relationship between the geometric model and the rule behavior of the physical manipulator, the tightening gun, the assembly equipment data, and the virtual model was constructed.

### 3.3. Virtual–Real Interaction

Virtual–real interaction is the bridge between virtual assembly line and physical assembly line. The realization of virtual–real interaction includes three steps: Firstly, with the purpose of realizing the interaction between virtual line and physical line, all kinds of data of physical assembly line are collected. Radio frequency identification, wireless sensor network, intelligent instruments, and various sensors are used to collect physical assembly line data and supervise the producing process. Secondly, based on the collected assembly line data, a high-speed and stable customized data transmission protocol is designed to synchronize and integrate data. When the assembly line is abnormal, such as abnormal bolt assembly torque, abnormal data will be sent to the virtual assembly line for abnormal data analysis and positioning visualization. Finally, because of the function of the digital twin model with simulation verification, the initial condition data of different working conditions are simulated to obtain the parameters in the operation process, and the optimal parameters are obtained by optimization. 

## 4. Torque Prediction Model Based on GWO-VMD-LSTM

### 4.1. Variational Modal Decomposition

Variational mode decomposition (VMD) is a completely non-recursive variational mode decomposition method [36], which can obtain the optimal center frequency and finite bandwidth for every mode in the decomposition process. Compared with the empirical mode decomposition (EMD) method [37], VMD has better decomposition performance for strong random and non-stationary signals, and the decomposed subsequences are more regular. However, the drawback of VMD is that the total models *K* and penalty factor *α* need to be manually set. The optimal *K* and *α* can be obtained by gray wolf optimization algorithm and permutation entropy in this paper. This part will be described in Section 4.3.

The non-stationary data *X*(*t*) is decomposed into *K_th_* intrinsic mode functions (*IMFs*) by VMD. Each modal component *u_k_* (*k* = 1, 2,…, *K*) corresponds to a certain central frequency *ω_k_*. In the decomposition process, the sum of all modes is equal to the original data *f*(*t*). The expression of the VMD is listed as:(1)$$\{\begin{array}{c} \underset{\left\{u_{k}\},\{\omega_{k}\right\}}{\min}\sum_{k=1}^{K}\left\{{\left\|\sigma_{t}[(\delta (t)+\frac{j}{\pi t})*u_{k}(t)]e^{-j\omega_{k}t}\right\|}_{2}^{2}\right\}\\ \;s.t.\sum_{k=1}^{K}u_{k}=f(t) \end{array}$$

In this equation, *f*(*t*) is the original data, *δ*(*t*) means the unit pulse function, *t* represents timestamp, “*” is convolution operation, *u_k_* and *ω_k_* represent the *k*th modal component and the corresponding central frequency, respectively, and *K* is the total number of modes. *A* and *λ*(*t*) are introduced to transform constraint problem Equation (1) into unconstrained problem:(2)$$\begin{array}{r} L(\{u_{k}\},\{\omega_{k}\},\lambda )=\alpha \overset{K}{\underset{k=1}{\sum }}∥\sigma_{t}[(\delta (t)+\frac{j}{\pi t})*u_{k}(t)]e^{-j\omega_{k}t}{∥}_{2}^{2}+∥f(t)-\\ \overset{K}{\underset{k=1}{\sum }}u_{k}(t){∥}_{2}^{2}+\left[\lambda (t),f(t)-\overset{K}{\underset{k=1}{\sum }}u_{k}(t)\right] \end{array}$$

In the equation, *λ*(*t*) is a Lagrangian operator, *α* is a quadratic penalty factor and $||f(t)\;-\overset{K}{\underset{k=1}{\sum }}u_{k}{\left(t)|\right|}_{2}^{2}$ is a quadratic penalty term. Equation (2) can be solved by the alternating multiplier optimization method to acquire the saddle point of the augmented Lagrange equation, and *u_k_* and *ω_k_* is denoted as follows:(3)$$u_{k}^{n+1}=\arg\min_{u_{k}}(\{u_{i}^{n+1}\},\{u_{i\geq k}^{n+1}\},\{\omega_{i}^{n}\},\{\lambda^{n}\})$$
(4)$$\omega_{k}^{n+1}=\arg\min_{\omega_{k}}(\{u_{i}^{n+1}\},\{\omega_{i<k}^{n+1}\},\{\omega_{i\geq k}^{n}\},\{\lambda^{n}\})$$
(5)$$\lambda^{n+1}=\lambda^{n}+\tau (f(t-\overset{K}{\underset{k=1}{\sum }}u_{k}^{n+1}))$$

The convergence condition is:(6)$$\overset{K}{\underset{k=1}{\sum }}∥u_{k}^{n+1}-u_{k}^{n}{∥}_{2}^{2}/∥u_{k}^{n}{∥}_{2}^{2}<\varepsilon$$
where, *n* is the iteration time, *τ* is the noise tolerance, *ε* is the given discriminant condition, when the signal contains noise, *τ* can filter noise. After calculation, *u_k_* and *ω_k_* are obtained as follows:(7)$$u_{k}^{n+1}(\omega )=\frac{f(\omega )-\sum_{i\neq k}u_{i}(\omega )+\frac{\lambda (\omega )}{2}}{1+2\alpha (\omega -\omega_{k})}$$
(8)$$\omega_{k}^{n+1}=\frac{\int_{0}^{\infty }\omega |u_{k}^{n+1}(\omega ){|}^{2}d\omega }{\int_{0}^{\infty }|u_{k}^{n+1}(\omega ){|}^{2}d\omega }$$
where, ${f(\omega ),\;\lambda (\omega ),\;u}_{i}{(\omega ),\;u}_{k}^{n+1}\left(\omega \right)$ is the Fourier transform of ${f(t),\;\lambda (t),\;u}_{i}{(t),\;u}_{k}^{n+1}\left(t\right)$, respectively.

The specific steps of VMD decomposition algorithm are listed as:
Initialize ${\{u}_{k}^{1}{\},\;\{\omega }_{k}^{1}{\},\lambda }^{1},\;n\leftarrow 0$;*n* = *n* + 1, performing the whole cycle;*k* = *k* + 1, until *k* = *K*, for all *ω* ≥ 0, *λ^n^*
^+ 1^, *u_k_*, and *ω_k_* are updated by Equations (3)–(5);Steps 2 and 3 of the cycle, when the Equation (6) is satisfied, the cycle ends.

### 4.2. Gray Wolf Optimization Algorithm

Gray wolf optimization algorithm (GWO) is a new metaheuristic method [38] proposed by Mirjalilietal in 2014. The algorithm is widely used in the optimization of algorithm parameters because of its simple structure, few adjusted parameters, and fast convergence speed. GWO classifies the initial population into *α*, *β*, *δ*, and *ω* gray wolves of different grades, and assigns the tasks of encirclement, hunting, and attack to gray wolves of different grades. Among them, *α*, *β*, *δ* are the best wolves, leading other wolves *ω* to find a better location for predation, and realizing the global optimization process. Figure 3 shows that *α*, *β*, *δ* estimate the range of prey, and then the locations of other wolves nearby are constantly updated. Finally, the wolf attacks and obtains the prey determine the optimal results. When the *α* value is approximately 0, the next update position of the wolf will be closer to the prey, namely, the optimal solution.

In this study, the *K* and *α* in VMD is optimized by GWO algorithm. The optimal solution vector is the optimal parameters *K* and *α*. Section 4.3 will introduce the specific optimization process of parameters (*K*, *α*) in detail.

### 4.3. GWO Optimized VMD Decomposition Method

In this section, GWO algorithm is proposed to optimize the VMD decomposition parameters (*K*, *α*), so that it can adaptively decompose the time series data. We name it the VMD based on GWO optimization method (GWO-VMD). The ratio of mean and variance of each subsequence permutation entropy (PE) after VMD decomposition is used as the fitness function of GWO-VMD algorithm. PE was proposed by Bandt et al. [39] in 2002, and it has been widely used in data analysis algorithms to evaluate temporal information embedded in time series. PE can reflect the law of time series data based on Shannon entropy of ordered pattern distribution in time series. The smaller the permutation entropy, the stronger regularity of the time series data.

If the original time series data is decomposed into *IMF_k_* (*k*= 1, 2, …, *K*) by VMD, the calculation steps of the fitness function based on PE are as follows:Phase space reconstruction

The univariate time series is expressed as {*x_t_*, *t* = 1, 2,…, *n*}, and the phase space is reconstructed to acquire:(9)$$X_{j}=[x_{j},x_{j+\hat{\tau }},\ldots ,x_{j+(\hat{m}-1)\hat{\tau }}]$$

Among them, $\hat{m}$ is the embedding dimension, $\hat{\tau }$ is the delay time, $\hat{m}$ and $\hat{\tau }$ are positive integers, *X_j_* represents the reconstruction vector *j* = 1, 2 …, n − $(\hat{m}-1)\hat{\tau }$.

2.The reconstructed vector ascending order arrangement

The elements of the reconstructed vector *X_j_* are arranged in ascending order to obtain:(10)$$x_{j+(j_{1}-1)\hat{\tau }}\leq x_{j+(j_{2}-1)\hat{\tau }}\leq \ldots \leq x_{j+(j\hat{m}-1)\hat{\tau }}$$

3.Reconstructing the vector *X_j_* into a sequence of symbols

Mapping the reconstructed vector to a symbol sequence based on the permutation symbol of Equation (10).
(11)$$S(g)=(j_{1},j_{2},\ldots ,j_{\hat{m}}),g=1,2,\ldots ,m!$$

4.Calculate the permutation entropy

Each symbolic sequence probability is {*P*(*g*), *g* = 1, 2,…, *m*!}. The PE of the *IMF_k_* is:(12)$$H_{p}(IMF_{k})=-\overset{m!}{\underset{i=1}{\sum }}P_{i}(IMF_{k})\ln(P_{i}(IMF_{k})),k=1,2,\ldots ,K$$

5.Selection and calculation of fitness function

After different (*K*, α) decomposition, the PE of each *IMF* may fluctuate. The ratio of the mean and variance of the permutation entropy is used as the fitness, and the fitness function goal is to obtain the minimum fitness value, as shown in the following equation:(13)$$fit_{mv}=\frac{mean(H_{p}(IMF_{k}))}{\mathrm{var}(H_{p}(IMF_{k}))}(k=1,2,\ldots ,K)$$

The flow chart of time series decomposition method based on GWO-VMD is shown in Figure 4. The specific steps are as follows:Input the original time series signal *x_t_*, initialize the gray wolf population as *N*, and the maximum iterations is *T_max_*. Set the value range of VMD parameter pair (*K*, *α*);The original time series *x_t_* is decomposed by VMD using initialized *K* and *α*, and the fitness value *fit_mv_* of all subsequences is calculated according to Equation (13), and retained as the current minimum fitness value;*fit_mvt_* of the current iteration is compared with the optimal fitness value *fit_mvbest_* of the previous iteration. If *fit_mvt_* < *fit_mvbest_*, then the current optimal fitness *fit_mvbest_* = *fit_mvt_*;Update the position of Wolf *α*, *β* and *δ*;Cycle iteration step 3 and step 4, until *T_max_* is reached, then the minimum fitness value of GWO and the optimal parameters (*K*, *α*) are obtained, and the VMD parameter optimization process is completed.

### 4.4. Long Short-Term Memory Network

The long short-term memory network (LSTM) is raised to solve the long-term data dependence problem existing in the recurrent neural network (RNN). The LSTM increases the forgetting gate, the input gate, the output gate, and the unit state. By training the weight parameters and the offset parameters of the model, the “gradient disappearance” and other problems existing in the RNN network can be avoided when the model parameters are unchanged. The Forgetting Gate may determine the extent of information retained at the previous moment. Through the three gates structure, the network connects the time relations before and after the time series data, so that the model can complete the selection of the previous data characteristics and complete the prediction of the next time step and longer time data. Figure 5 shows the detail of LSTM network.

Forget gate

The forgetting gate can calculate the probability of the last moment being forgotten using the sigmoid function. The obtained value is multiplied by the state of the hidden layer at the previous moment to determine the value retained at the previous moment. The calculation equation of forget gate is listed as follows:(14)$$f_{t}=\sigma (W_{f}\cdot x_{t}+W_{f}\cdot h_{t-1}+b_{f})$$
(15)$$\sigma (x)=\frac{1}{1+e^{-x}}$$

2.Input gate

The input gate multiplied the *i_t_* updated by the sigmoid function and the information ${\tilde{C}}_{t}$ updated by the tanh activation function to update the state of the unit. The calculation equation of the input door is denoted as follows:(16)$$i_{t}=\sigma (W_{i}\cdot x_{t}+W_{i}\cdot h_{t-1}+b_{i})$$
(17)$${\tilde{C}}_{t}=\tanh(W_{c}\cdot x_{t}+W_{c}\cdot h_{t-1}+b_{c})$$

*W_i_*, *W_c_* are the weight coefficient; *b_i_*, *b_c_* are bias coefficients; *f_t_* is the output of the forgetting gate, which controls the degree of the cell state *C_t_*_−1_ in the upper layer is forgotten, and $i_{t}\cdot {\tilde{C}}_{t}$ represents the new information retained in this layer. The current layer cell state *C_t_* update equation is:(18)$$C_{t}=f_{t}\cdot C_{t-1}+i_{t}\cdot {\tilde{C}}_{t}$$

3.Output gate

The output gate is used to control the number of filtered cell states in this layer. The value of *o_t_* is obtained by using the sigmoid activation function. The tanh activation function is used to process the cell state *C_t_* and multiply with *o_t_* to obtain the output *h_t_* of this layer. The equation for calculating the output gate is denoted as:(19)$$o_{t}=\sigma (W_{o}\cdot x_{t}+W_{o}\cdot h_{t-1}+b_{o})$$
(20)$$h_{t}=o_{t}\cdot \tanh(C_{t})$$

*o_t_* is the output; *h_t_* represents the final output; *w_o_* denotes the weight coefficient; *b_o_* represents the bias coefficient.

### 4.5. Proposed GWO-VMD-LSTM Torque Prediction Model

The assembly torque sequence of the automobile rear axle presents the characteristics of randomness, volatility, and discontinuity. The GWO-VMD decomposition algorithm proposed in Section 3.2 can decompose the original assembly torque sequence data into a series of continuous and stable modal *IM**Fs*. Therefore, in this section, a time series prediction algorithm combining with GWO-VMD and LSTM is raised to solve the assembly torque prediction problem of the rear axle. As shown in Figure 6, the workflow of the proposed prediction algorithm is constructed.

Torque prediction modeling process is listed as follows:GWO-VMD modal decomposition. The GWO-VMD method is used to perform the modal decomposition of the original sequence of assembly torque to obtain k subsequences (*IMF*_1_,…, *IMF_K_*) as well as the residual term *res*;The *k*th sequence {*x_k1_*, *x_k2_*,…, *x_kn_*} is selected, where *x_kt_* is the sequence value at the moment *t* of the *k* sequence, *t* = 1, 2, …, *n*;To reconstruct the input–output sample pair for the *k*th sequence, set the sequence {*x_kt_*} obtained from step 2 to the initial embedding dimension *m*. Perform phase space reconstruction to establish the mapping *f: R^m^→R*, ${\hat{x}}_{kt}$ = *f*(*x*_*k*(*t* − (*m* − 1))_,…, *x*_*k*(*t* − 1)_, *x_kt_*). The specific input–output sample pair is:(21)$$X_{k}=\left[\begin{array}{ccc} x_{k1} & \ldots & x_{km}\\ \vdots & \ddots & \vdots \\ x_{k(n-m)} & \ldots & x_{k(n-1)} \end{array}\right],Y_{k}=\left[\begin{array}{c} x_{k(m+1)}\\ \vdots \\ x_{kn} \end{array}\right]$$

*X_k_* is the input matrix of the *k*th sequence sample and *Y_k_* is the output matrix of the *k*th sequence sample;

4.The LSTM prediction model is established for each subsequence and item, and the sample pairs obtained in step 3 are trained as training sets and test sets to obtain the prediction model;5.Prediction. The trained subsequence *IMFs* and *res* prediction models are used to predict;6.Summation of prediction results. The predicted value of the original sequence is obtained by summing the prediction results of each subsequence and the residual term.

## 5. Experimental Results and Comparative Analysis

### 5.1. Data Preparation and Development Environment

This experiment studies the assembly process of a certain type of rear axle, and selects six sample data for rear axle brake caliper bolt assembly, named as Dataset-1~Dataset-6. Each sample is divided 8:2 into training set and test set. The GWO proposed in Section 4.3 is used to optimize the decomposition order *K* and *α* in the VMD. The iteration of GWO is set to 100, the initial population size is 25, and the variable dimension is set to 2. Table 1 provides the detailed classification numbers of the six datasets and the VMD decomposition parameters *K* and *α* after GWO optimization. The development environment configuration in this work is: Python 3.8 (Guido van Rossum, Amsterdam, The Netherlands), tensorflow1.8 (Google, Mountain View, CA, USA), 2 GB Intel HD graphics GPU (Microsoft, Redmond, WA, USA), and Intel i7-7560U CPU(Microsoft, Redmond, WA, USA).

### 5.2. Evaluation Index

We often use the evaluation index of mean absolute error (*MAE*), root mean square error (*RMSE*), mean absolute percentage error (*MAPE*) and deterministic to characterize the performance of algorithm models. The evaluation indexes are listed as follows:(22)$$MAE=\frac{1}{n}\overset{n}{\underset{t=1}{\sum }}|(x_{t}-{\hat{x}}_{t})|$$
(23)$$RMSE=\sqrt{\frac{1}{n}\overset{n}{\underset{t=1}{\sum }}{\left(x_{t}-{\hat{x}}_{t}\right)}^{2}}$$
(24)$$MAPE=\frac{100\%}{n}\overset{n}{\underset{t=1}{\sum }}|\frac{{\hat{x}}_{t}-x_{t}}{x_{t}}|$$
(25)$$R^{2}=1-\frac{\sum_{t=1}^{n}{\left({\hat{x}}_{t}-x_{t}\right)}^{2}}{\sum_{t=1}^{n}{\left({\bar{x}}_{t}-x_{t}\right)}^{2}}$$

In the above formula, *x_t_* and ${\hat{x}}_{t}$ are the true value and forecast value at time *t*, respectively, *n* is the length of samples. *MAE*, *RMSE*, and *MAPE* mainly reflect the forecast error. The larger the error value, the worse the forecast accuracy. *R*^2^ indicates the proximity between predicted and observed values, and the closer *R*^2^ to 1, the better forecast performance.

### 5.3. Assembly Torque Prediction

The GWO-VMD-LSTM model utilized the first 10 sets of assembly torque data arranged in time series to predict the 11th set of assembly torque data. The original torque data sequence was firstly decomposed into *K* data subsequences by the GWO-VMD algorithm, and then the prediction model of each subsequence was established on the basis of LSTM neural network. The parameters of each LSTM were set as follows: the epoch is 300, the number of hidden layer neurons is 6, and the batch size is 1. In addition, the activation function used by the input and output gates is “tanh”, and the “Adam” optimizer uses *RMSE* as the training loss function. Finally, each data subsequence was predicted by using the trained model, and then the prediction results of the original torque data sequence can be obtained by summing the forecast results of each subsequence. As shown in Figure 7, the predicted and actual values of each *IMF* were calculated after Dataset-3 was decomposed by VMD. According to Figure 7a–h, the LSTM model can accurately predict *IMFs* of each data subsequence with stable fluctuations, but the residual sequence with high frequency (as shown in Figure 7i) has a poor prediction effect, which also shows that non-stationary data is difficult to predict specialty.

With the purpose to verify the predictive performance of the GWO-VMD-LSTM model proposed in this paper, the model was tested by using six datasets. Figure 8 shows the forecast effect of the proposed model with different datasets. From the prediction errors of the six datasets based on the model, which are depicted in Table 2, Dataset-1 and Dataset-2 have abnormal mutation data, so the *MAE* is larger than the other four groups. Dataset-3, Dataset-4, Dataset-5, and Dataset-6 have no abnormality and are relatively stable, so the prediction results are better. Thus, the decomposition prediction model based on GWO-VMD-LSTM proposed in this paper has good generalization ability under different datasets.

### 5.4. Comparative Analysis of Prediction Models

To further verify the property of the proposed GWO-VMD-LSTM model, the models of SVM, RNN, LSTM, GWO-VMD-SVM, and GWO-VMD-RNN are used to compare the prediction effect. As shown in Figure 9 and Figure 10, the prediction effect of the “decomposition-integration” model based on GWO-VMD is better than that of the “direct prediction” model, which indicates that the “decomposition-integration” method can predict extremely complex, highly non-stationary time series data.

According to the error evaluation indicators of different models, which are shown in Table 3 and Figure 11, the accuracy of the “direct prediction” method is extremely poor, and its *MAE* and *RMSE* are relatively large. The determination coefficient *R*^2^ of SVM and RNN is close to 0, indicating that the “direct prediction” method cannot meet the demand of assembly torque sequence prediction. Compared to the “direct prediction” model, the *MAPE* and *RMSE* of the “decomposition-integration” model are significantly reduced. Compared to the models based on GWO-VMD-SVM, GWO-VMD-RNN, SVM, RNN, and LSTM, the *RMSE* of the GWO-VMD-LSTM model is reduced by 61.5%, 57%, 82.9%, 83%, and 81.1%, respectively. Similarly, compared to the models based on GWO-VMD-SVM, GWO-VMD-RNN, SVM, RNN, and LSTM, the *MAE* of the GWO-VMD-LSTM model is reduced by 61.5%, 57.6%, 83%, 83.2%, and 81.1%, respectively. According to the experimental results, the GWO-VMD-LSTM model has the best performance and the highest accuracy among the comparison algorithms, so the model can meet the demand of the torque prediction.

## 6. Online Control of Assembly Torque Based on Digital Twin

### 6.1. Digital Twin System Development

The digital twin system for assembly line can improve the intelligent level of assembly line, realize the visualization of assembly line state and assembly process, and alarm the abnormal assembly torque. Catia was used for geometric modeling of assembly line, and the virtual model scene was constructed and rendered on unity platform. Finally, C# and IoTDB timing database were used to realize real-time synchronization between virtual space and physical space by using virtual–real synchronous interface. The synchronous response time of the system was less than 1 s. According to the GWO-VMD-LSTM assembly torque prediction algorithm model constructed in Section 4, the rear axle assembly torque prediction is realized. Through the feedback control module of digital twin, the online control of rear axle assembly torque is realized. In Figure 12, the visualization window can present the state and assembly process of the current physical production line in real time from the global and station mode. The data supervising display window can supervise the torque value, predicted torque value, torsion angle, and other parameters collected in real time.

### 6.2. On-Line Control of Assembly Torque

Based on the torque prediction model proposed in Section 4.5, the assembly torque can be accurately predicted, and the online control of torque can be realized through the decision module, which is great important to the assembly quality control of the rear axle. If the torque value is not within the preset threshold range, the drive motor parameters (motor speed and output torque) need to be adjusted to design range. This is a highly dynamic process, which requires an effective method to realize real-time interactive fusion of assembly process information. Digital twin is a virtual environment, which truly reflects the working state of the physical assembly line by sensor updating and historical data. Using digital twin technology, the real-time prediction of assembly torque can be realized, and the synchronous adjustment of motor output torque can be realized by decision module. As shown in Figure 13, we propose a rear axle assembly torque online control method found on digital twin. The validity and accuracy of the constructed torque prediction model has been verified in Section 5.1, Section 5.2 and Section 5.3. Therefore, it can be coupled to the digital twin system and predict assembly torque in real time. The real-time dynamic data collected include assembly torque, torsional angle, manipulator pose, tightening gun pose, etc. These data are used to model and predict torque in virtual. At the same time, the decision-making module updates the equipment parameters according to the predicted torque, and feeds back to the motor of assembly line for controlling the output torque of the tightening gun. The complete online control method steps are listed: Firstly, the initial assembly determined parameters.The assembly torque is measured in real-time, and the torque prediction model is used to forecast the next moment assembly torque.If the predicted torque is within the set threshold range, it means that the current assembly parameters satisfy the assembly requirements and continue to assemble with the designed output torque; if the forecast value is abnormal, it is necessary to adjust the torque value and the position of the tightening gun, and generate instructions to the drive motor until the sensor returns the normal torque value.The whole process is repeated until the assembly is completed.

## 7. Conclusions

In this paper, a rear axle assembly torque online control method combined with digital twin was developed to improve the quality and efficiency of rear axle assembly. The main conclusions are as follows:We proposed an assembly torque prediction model based on GWO-VMD-LSTM algorithm, and the prediction accuracy of the model reaches 99.49%.SVM, RNN, LSTM, GWO-VMD-SVM, GWO-VMD-RNN, and GWO-VMD-LSTM are compared, and the corresponding *R*^2^ indexes are 0.09, 0.085, 0.257, 0.822, 0.855, and 0.974, respectively, which demonstrated that the GWO-VMD-LSTM algorithm is the best.The digital twin system of rear axle assembly line was constructed, and the system’s response time reaches 1 s, according to experimental test, which can realize the online control of assembly torque.

The developed digital twin system can realize real-time supervising of assembly line state and the visualization of assembly process. Based on the work above, optimizing the algorithm parameters proposed in this paper and comparing with the frontier algorithms will further improve the forecast property of the proposed algorithm in this paper. This work will be systematically studied in future work.

## Figures and Tables

**Figure 1 sensors-22-07282-f001:**
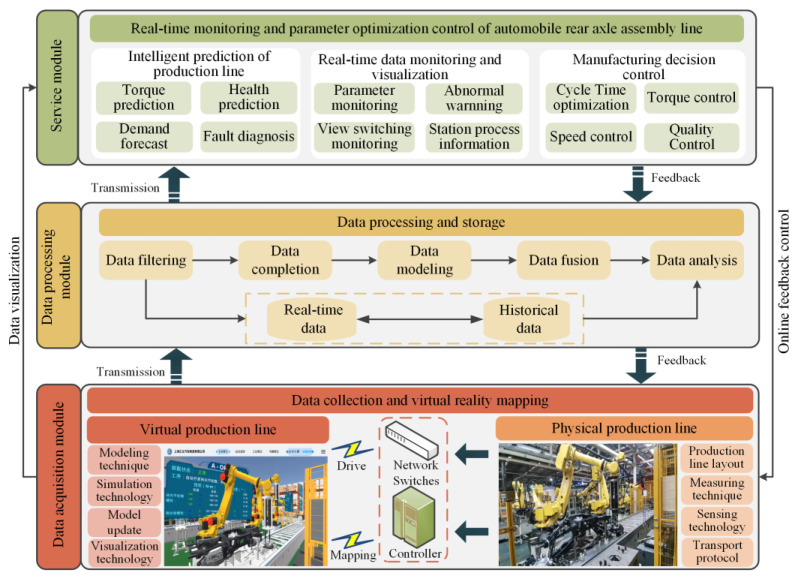
Overall architecture of the rear axle assembly line digital twin system.

**Figure 2 sensors-22-07282-f002:**
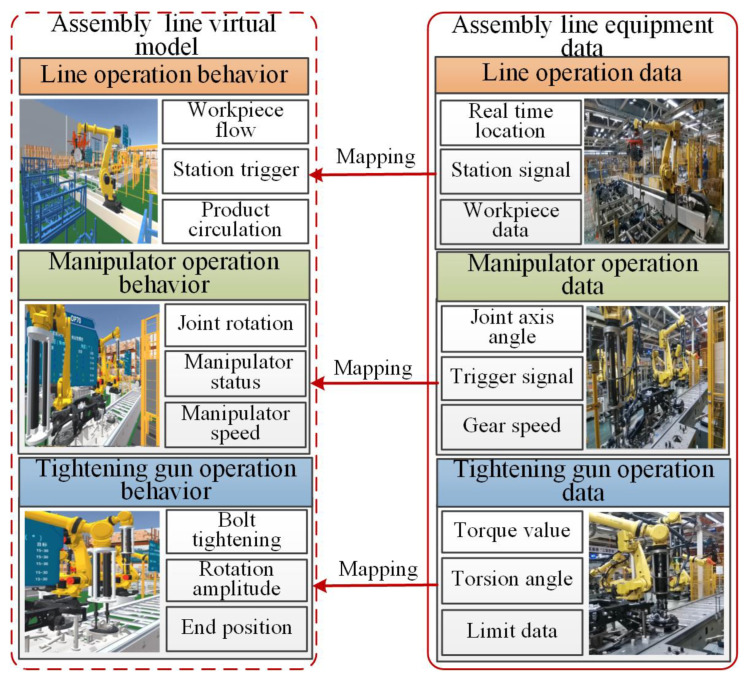
Mapping relationship between data and twin Models.

**Figure 3 sensors-22-07282-f003:**
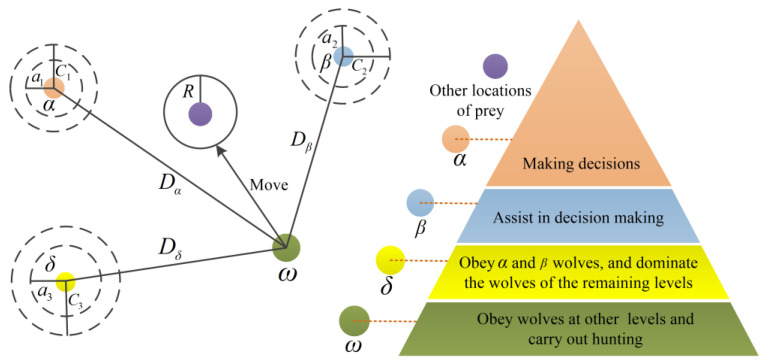
The hierarchical structure of wolves in the GWO algorithm.

**Figure 4 sensors-22-07282-f004:**
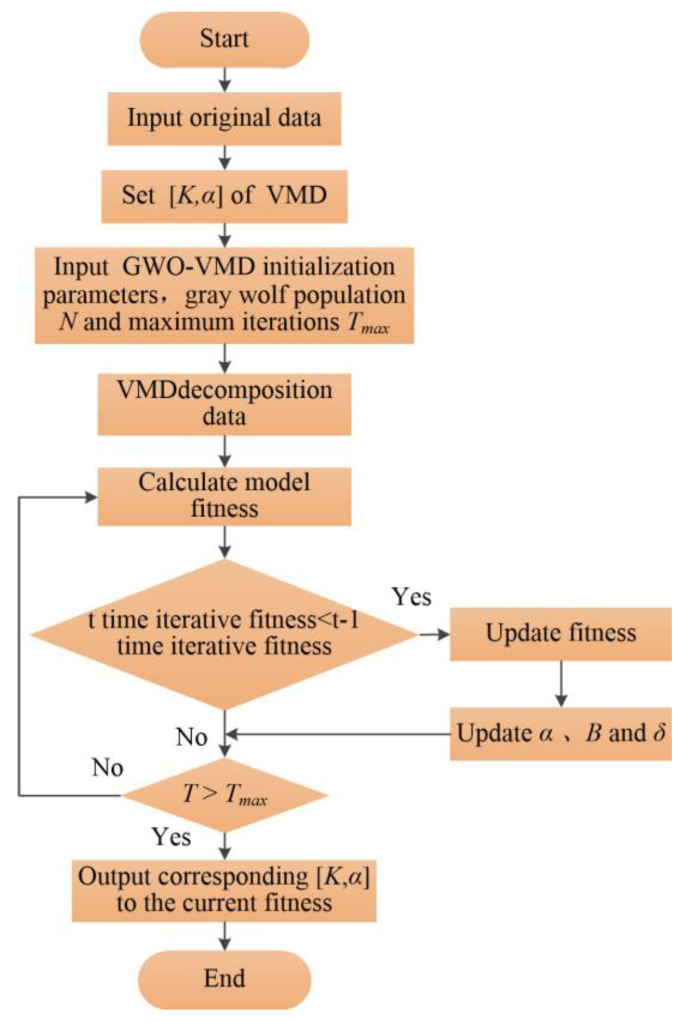
GWO−VMD time series decomposition method.

**Figure 5 sensors-22-07282-f005:**
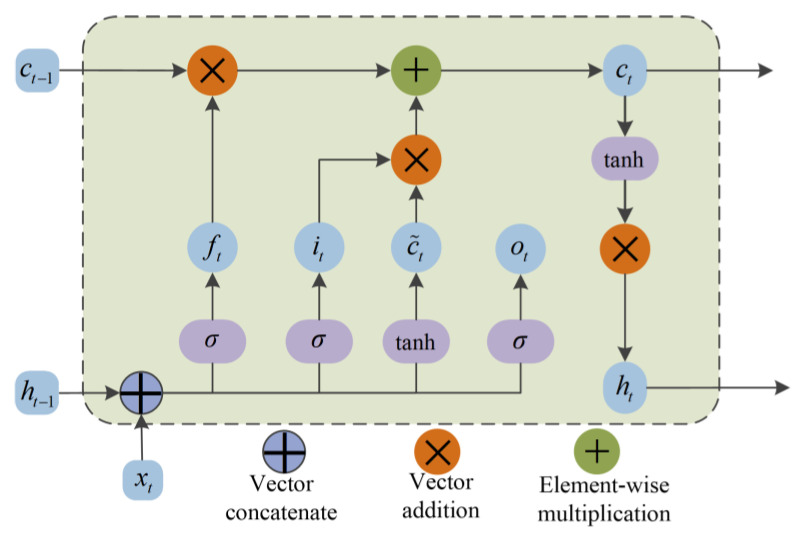
LSTM network structure diagram.

**Figure 6 sensors-22-07282-f006:**
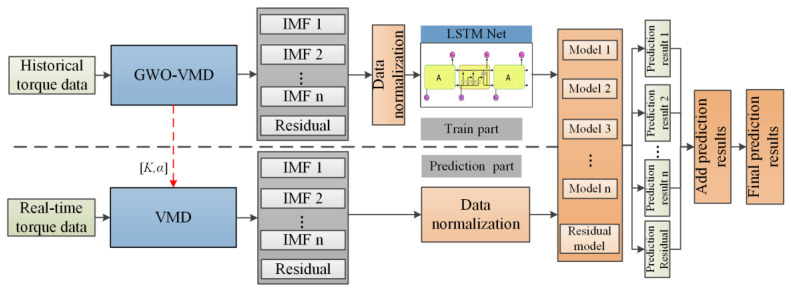
GWO-VMD -LSTM prediction model.

**Figure 7 sensors-22-07282-f007:**
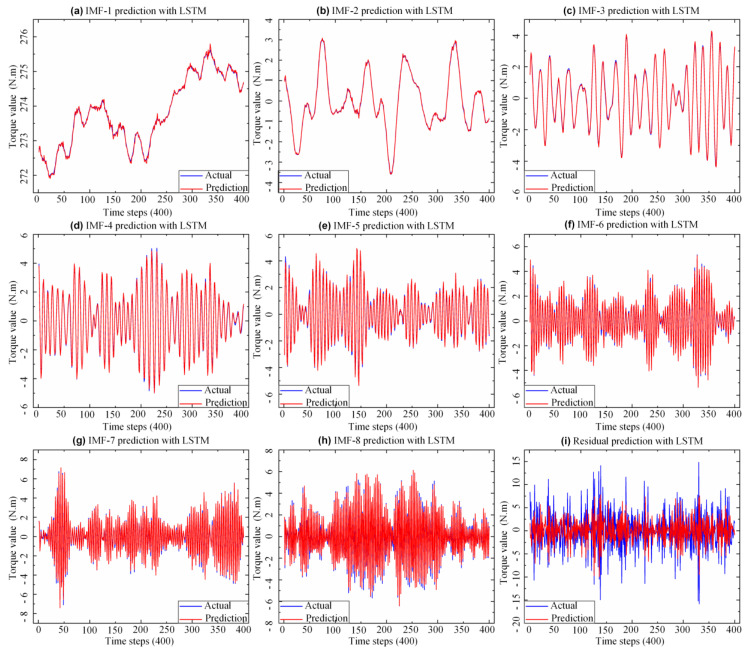
GWO−VMD decomposition *IMFs* prediction results.

**Figure 8 sensors-22-07282-f008:**
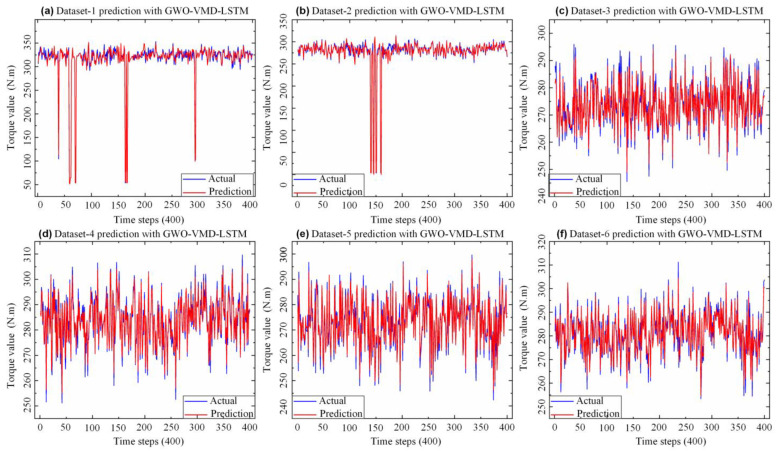
Dataset−1~6 prediction results.

**Figure 9 sensors-22-07282-f009:**
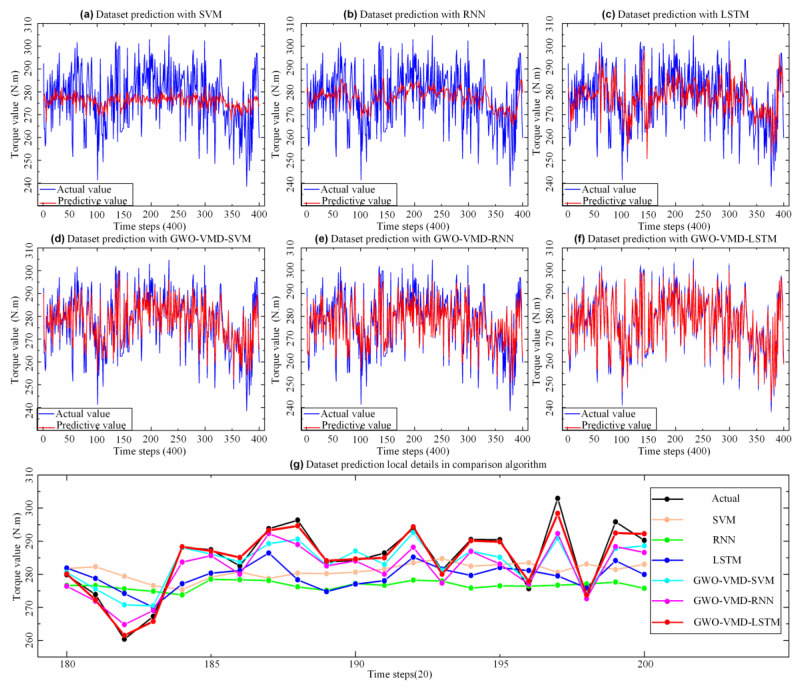
Prediction effects of different models on selected datasets.

**Figure 10 sensors-22-07282-f010:**
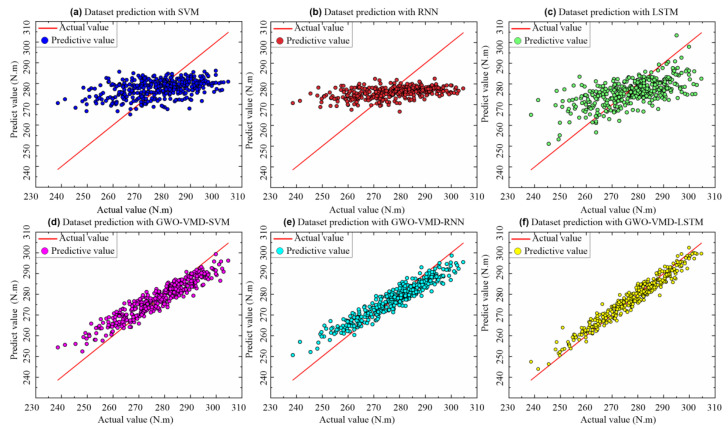
Fitting results of true and predicted values of different models.

**Figure 11 sensors-22-07282-f011:**
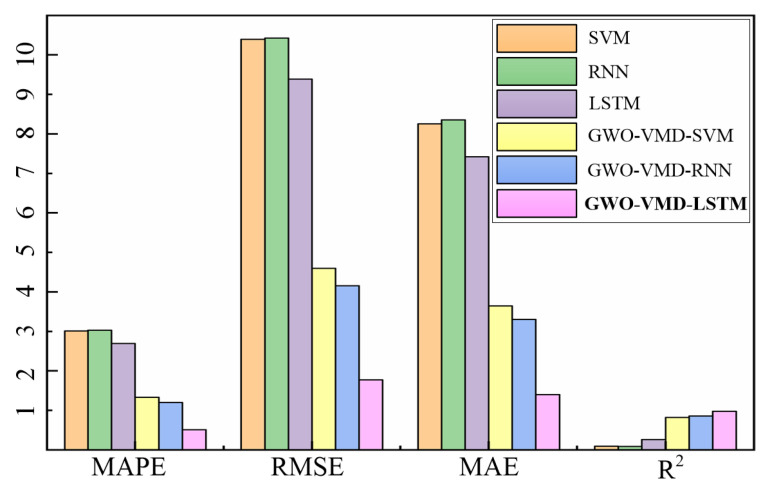
Comparison of prediction error between different models for selected data sets.

**Figure 12 sensors-22-07282-f012:**
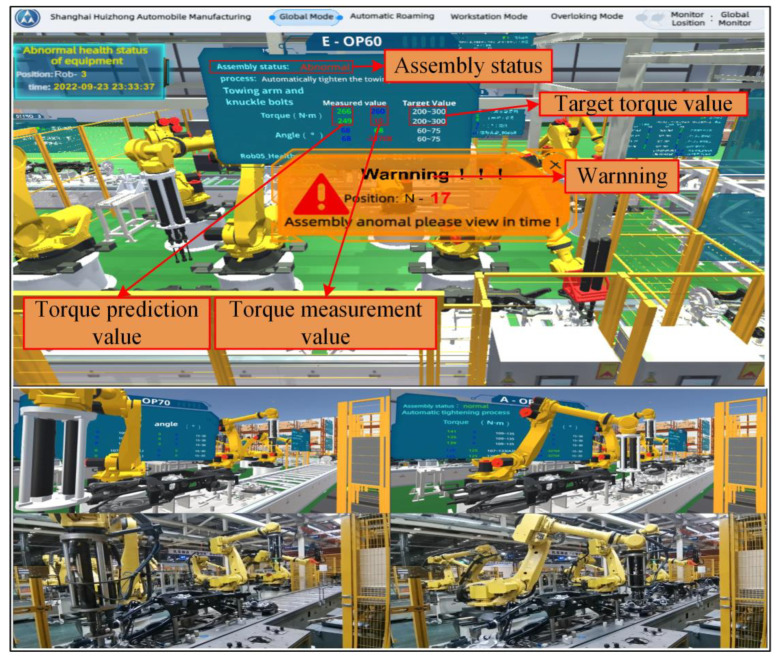
Digital twin system for rear axle assembly line.

**Figure 13 sensors-22-07282-f013:**
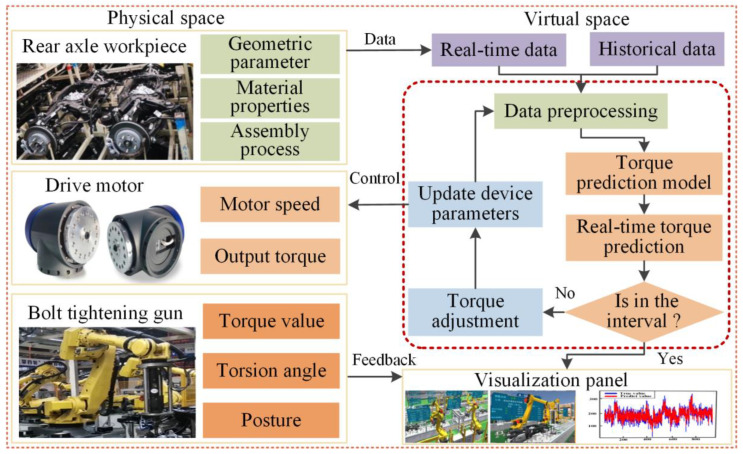
Online torque control based on digital twin assembly.

**Table 1 sensors-22-07282-t001:** Datasets and VMD decomposition parameters.

Index	Total Number	Train Number	Test Number	VMD Parameter	
				*K*	*α*
Dataset-1	200	1600	400	11	1861
Dataset-2	2200	1760	440	4	2776
Dataset-3	2100	1680	420	8	2725
Dataset-4	2000	1600	400	11	2956
Dataset-5	2110	1688	422	4	1209
Dataset-6	2120	1696	424	9	2700

**Table 2 sensors-22-07282-t002:** GWO-VMD-LSTM prediction error of six datasets.

Datasets	*MAPE* (%)	*RMSE*	*MAE*	*R* ^2^
Dataset-1	9.87	8.765	6.917	0.806
Dataset-2	9.37	8.628	5.927	0.824
Dataset-3	0.58	2.070	1.590	0.947
Dataset-4	0.58	2.042	1.630	0.958
Dataset-5	0.66	2.237	1.785	0.951
Dataset-6	0.64	2.238	1.781	0.943

**Table 3 sensors-22-07282-t003:** Performance prediction for each model of selected dataset.

Model	*MAPE* (%)	*RMSE*	*MAE*	*R* ^2^
SVM	3.01	10.393	8.256	0.09
RNN	3.92	10.423	8.353	0.085
LSTM	2.69	9.389	7.419	0.257
GWO-VMD-SVM	1.33	4.597	3.647	0.822
GWO-VMD-RNN	1.20	4.155	3.303	0.855
GWO-VMD-LSTM	0.51	1.770	1.400	0.974

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
