# Peer review of "Digital Twin-Driven Rear Axle Assembly Torque Prediction and Online Control"

_sensors, 2022, doi:10.3390/s22197282_

Round 1

Reviewer 1 Report

(1) The authors need to summarize the disadvantages of existing methods and the specific problems mainly to be solved in this paper in section 1.

(2) Some variable symbols in Eq. (1) are missing.

(3) Please list the reasons for choosing VMD to pre-process the data. Does this method have any significant advantages in this field?

(4) There are several boxes for "Prediction result 1" in Figure 6, is it a labeling error or these results are originally same?

(5) The prediction errors in Figure 8 for dataset 1 and 2 are in the same position (around 150) . It seems that this phenomenon is not random, and it will be more persuasive if  this error can be solved. In addition, the blue and red line in the legend of Figure 8(a) are missing.

Reviewer 2 Report

This manuscript addresses a problem of assembly torque prediction in manufacturing processes. It outlines the main problem along with the lack of concrete methods to solving it due to the lack of known parameters in real applications. The authors suggest creating a digital twin model of the assembly line which is used to evaluate the required torque and adjust control parameters of the manufacturing line in real time. The experiments show that the delay between the analysis and feedback control is around 1s which is acceptable. The authors go into great details discussing the structure of digital twin model – data acquisition and processing platforms – and propose a machine learning-based system for torque prediction. They compare their combined algorithm with other conventional and ML techniques showing that the proposed method performs the best reaching over 99% accuracy.

The paper is well-structured and very informative providing great overview of the research area as well as novel practical results. In the opinion of the reviewer, it can be published in its present form after final proofreading. The only critique the reviewer has is small dataset size collected for training and evaluating ML model accuracy. However, the reviewer understands that collecting data for such experiments is costly, so this minor problem can be ignored.

Reviewer 3 Report

In this paper, the authors proposed a rear axle assembly torque prediction model based on gray wolf optimized variational modal decomposition and long short-term memory network (GWO-VMD-LSTM). The proposed technique is a combination of well-known traditional techniques. Overall, the authors have made a good attempt. However, I do not recommend this paper to editors, because the similarity of this paper is too high. In iThenticate, the similarity of this manuscript is “38 %”. The reviewer’s other comments are as follows:

1.      In the abstract part, the novelty and key idea of the proposed method should be described. The authors only described that “First, a rear axle assembly torque prediction model based on gray wolf optimized variational modal decomposition and long short-term memory network (GWO-VMD-LSTM) was developed, …”. The novelty and key idea are not clear.

2.      The authors should not use acronym without explanation. All acronyms must be defined before use. For example, “PIO-SVM”, etc.

3.      The problem definition of this work is not clear. In the introduction part, the drawbacks of each conventional technique should be described one by one. The authors should emphasize the difference with other methods to clarify the position of this work further.

4.      The authors should improve the mathematical presentation. For example, what does the square stand for in Eq. (1)? Besides, the authors should unify the font size of all equations.

5.      Please unify the font style. In sentences/equations, mathematical expressions should be Italic font. (Some of them are Italic fonts and others are Roman font.) For example, see Eq. (3), (4), etc. Otherwise, readers will be confused.

6.      The effectiveness of this work is not clear. Through simulations/experiments, the authors must justify the effectiveness of the proposed method by comparing with the other latest methods. Several articles are discussed in the research survey. However, no comparison is shown with these techniques. Frankly speaking, the research survey and References are meaningless. The authors should show comparison data.

Round 2

Reviewer 3 Report

Thank you for submitting the revised version of the manuscript ID: sensors-1884879. The reviewer would like to pay tribute to the authors’ great work. In the revised version, most of the reviewer’s requests were met by the authors. However, the following biggest concern was not improved in the revised version. The similarity of the revised version is still high. In iThenticate, the similarity of this manuscript is “34 %”. (See the similarity report.) The reviewer cannot recommend this paper as it is.

>The first comment from the reviewer #3:

In this paper, the authors proposed a rear axle assembly torque prediction model based on gray wolf optimized variational modal decomposition and long short-term memory network (GWO-VMD-LSTM). The proposed technique is a combination of well-known traditional techniques. Overall, the authors have made a good attempt. However, I do not recommend this paper to editors, because the similarity of this paper is too high. In iThenticate, the similarity of this manuscript is “38 %”. The reviewer’s other comments are as follows:

----

>Comment #4 from the reviewer #3:

The authors should improve the mathematical presentation. For example, what does the square stand for in Eq. (1)? Besides, the authors should unify the font size of all equations.

>Authors’ reply for the comment #3:

We ensure that the problem will not occur next time. Also, we have unified the font size of all equations as suggested.

>Reviewer’s reply:

I understood. However, Eq. (1) still has editing problem. (The above-mentioned problem was not solved in the revised version.) Please improve it.
